# Epidemiological and Immune Profile Analysis of Italian Subjects with Endometriosis and Multiple Sclerosis

**DOI:** 10.3390/jcm12052043

**Published:** 2023-03-04

**Authors:** Brunella Zizolfi, Virginia Foreste, Simona Bonavita, Valentina Rubino, Giuseppina Ruggiero, Vincenzo Brescia Morra, Roberta Lanzillo, Antonio Carotenuto, Francesca Boscia, Maurizio Taglialatela, Maurizio Guida

**Affiliations:** 1Department of Public Health, University of Naples Federico II, 80131 Naples, Italy; 2Department of Neuroscience, Reproductive Sciences and Dentistry, School of Medicine, University of Naples Federico II, 80138 Naples, Italy; 3II Clinic of Neurology, Department of Advanced Medical and Surgical Sciences, University of Campania “Luigi Vanvitelli”, 80138 Naples, Italy; 4Department of Translational Medical Sciences, University of Naples Federico II, 80138 Naples, Italy

**Keywords:** endometriosis, multiple sclerosis, autoimmune disease, pathogenesis

## Abstract

Objective: To report for the first time an Italian epidemiological analysis of the prevalence of multiple sclerosis (MS) in patients with endometriosis (EMS), through the study of the endometriosis population of our referral center; to analyze the clinical profile and perform a laboratory analysis to examine the immune profile and the possible correlation to other autoimmune diseases of the enrolled patients. Methods: We evaluated 1652 women registered with EMS in the University of Naples Federico II and retrospectively searched patients with a co-diagnosis of MS. Clinical features of both conditions were recorded. Serum autoantibody and immune profiles were analyzed. Results: 9 out of 1652 patients presented a co-diagnosis of EMS and MS (9/1652 = 0.005%). Clinically, EMS and MS presented in mild forms. Hashimoto’s thyroiditis was found in two patients (2/9). Even if not statistically significant, a trend of variation in CD4- CD8 T lymphocytes and of B cells were found. Conclusion: Our findings suggest an increased risk of MS in women with EMS. However, large-scale prospective studies are needed.

## 1. Introduction

Endometriosis (EMS) is an estrogen-dependent inflammatory disorder that mainly affects women of reproductive age [1]. The pathogenesis, mechanism of morbidity and management of EMS are still under investigation [2]. In 1987, Gleicher et al. presented the theory that EMS may be an autoimmune disease, considering that most of the criteria for autoimmune diseases, including polyclonal B cell activation, immunological abnormalities in T and B cell functions, increased apoptosis, tissue damage and multi-organ involvement, were satisfied [3]. According to recent studies, there is growing evidence that immune factors can contribute to create a pro-inflammatory microenvironment that facilitates the persistence of EMS, allowing the endometrial implants to survive in ectopic sites [4,5]. 

In this view, due to the immunological profile, a potential link between EMS and other autoimmune disease has been postulated [6].

In the literature, different risks of association have been evaluated. According to Sinaii et al., women with EMS are at higher risk of other autoimmune diseases, particularly multiple sclerosis (MS, 7–24-fold), in comparison to the general population [7], while according to Nielsen et al., women with EMS are either at a modestly increased risk (20–60%) or at no unusual risk of MS, SLE and SS [8].

According to this evidence, although MS and EMS are clearly different in their phenotypes, it seems that they could share a common immune background [9].

Indeed, EMS and MS are characterized both by systemic inflammation and immune system dysregulation with an increased level of inflammatory Th1 to Th2 cells, as well as a concomitant dysregulation of interferon-gamma [10]. However, there is still limited evidence explaining the detailed molecular or immunological mechanisms shared by these two diseases [11].

The first objective of this study is to perform an epidemiological analysis to evaluate the prevalence of MS in a cohort of Italian patients with EMS (endometriosis population of our referral center); the second aim is to analyze clinical conditions in patients with both EMS and MS; the final step consists in a laboratory analysis to examine the immune profile and the possible correlation to other autoimmune diseases in patients with both EMS and MS.

## 2. Materials and Methods

Patients were retrospectively recruited from the database of the reference center for the diagnosis and treatment of EMS of the gynecological department of the University of Naples Federico II. 

The database contains all patients suffering from endometriosis that had requested an ambulatory access for a clinical control since 2012. Thus, it is composed of all endometriosis patients between 15 and 50 years old that have been referred to our center for a clinical control. From this database, patients with the following inclusion criteria were enrolled: age between 15 and 50 years old, and diagnosis of MS according to the revised McDonald criteria [12]. To diagnose and classify EMS, the American Fertility Society (AFS) score was used, a classification that has been accepted globally and has been widely used in recent years. According to this classification system, the stage of endometriosis was derived from a cumulative score. The weighted value system was scored and summed according to the size of the endometriotic lesions in the ovaries, peritoneum and fallopian tubes, and the severity of adhesion at each of the mentioned sites. The staging system was divided into four stages: I (1 to 5 points, minimal), II (6 to 15 points, mild), III (16 to 40 points, moderate) and IV (>40 points, severe) [13].

Patients who met the following criteria were excluded: malignancies, other severe intercurrent conditions, previous hystero-annessiectomy and premature ovarian failure. 

All patients enrolled were invited to participate in a prospective study to evaluate the clinical features of both diseases and to examine their autoimmune serological profile. The study was conducted following the Declaration of Helsinki (1975) and Good Clinical Practice guidelines. 

Before enrolment, the purpose of the study was clearly explained, and all patients received detailed information about the study procedure, to which they gave their consent. The information obtained was anonymized before analysis. 

All data on patients’ EMS (clinical symptoms, stage, obstetric and gynecological history and treatment) and MS (onset disability score assessed by the expanded disability status scale (EDSS), disease-modifying treatment and brain and spinal MRI) at the time of enrolment and after one year were also collected to understand how their personal therapy can control the diseases courses. A pharmacological anamnesis was conducted. Finally, a peripheral blood sample (4 mL) was taken to analyze the presence of autoantibodies (ANA, AMA, AGA, EMA, dsDNA, ASMA, APCA, ATA, anti-TG and anti-TPO) in the serum to exclude other immunological disorders and to evaluate the immune profile (CD4/CD8 ratio, percentage of B lymphocytes and percentage of NK and iNKT lymphocytes). FITC-, PE-, Pecy5- and Pecy7-labelled mAb against CD3, CD4, CD8, CD56, CD19, Va24, CD54 and isotype-matched controls were purchased from BD PharMingen (San Jose, CA, USA). One out of nine patients refused peripheral blood sampling. 

For the analysis of the immune profile, a group of 8 patients, aged between 15 and 50 years old, with only a diagnosis of EMS, were recruited as control group; patients who met the following criteria were excluded: malignancies, other severe intercurrent conditions, previous hystero-annessiectomy and premature ovarian failure (Table 1). Patients of the control and the study group were balanced for age, BMI, EMS stage and management. 

All phenotypes referred to flow cytometry analysis of the lymphocyte population gated using forward scatter (FSC) and side scatter (SSC) parameters. Flow cytometry and data analysis were performed using an ATTUNE NxT acoustic focusing cytometer (life technologies) and the analysis was performed by ATTUNE NxT software.Ink.

### Statistical Analysis

The statistical evaluation of the data, by InStat 3.0 software (GraphPad Software Inc., San Diego, CA, USA), was performed by means of the Mann–Whitney test. Two-sided *p* values of less than 0.05 were considered to indicate statistical significance.

## 3. Results

### 3.1. Epidemiological Analysis 

A total of 9 out of 1652 patients (9/1652 = 0.5%) had a co-diagnosis of EMS and MS. The general characteristics of patients are summarized in Table 2. 

### 3.2. Clinical Evaluation 

The recruited patients had mild forms of both diseases. Table 3 (EMS) and Table 4 (MS) summarize the characteristics described. 

In relation to EMS, one patient had undergone drug treatment (ended in 2018) and none had undergone surgery; at the one-year follow-up ultrasound, no significant progression was evidenced; regarding pregnancy, 3 out of 9 desired or had experienced pregnancy, 1 with an assisted reproduction technique, while 5 out 9 did not seek pregnancy.

In consideration of the neurological aspect, all patients had a relapsing remitting form of MS, with an average EDSS of 3.

The diagnosis of EMS followed the MS one in the whole group and in most cases, the diagnosis of EMS was fortuitous (Figure 1).

### 3.3. Immunological Pattern 

One out of nine patients refused peripheral blood sampling. Of the 8 screened patients, 2 had anti-thyroperoxidase autoantibodies (AbTPO). For the analysis of the immune profile, a group of 8 patients with only EMS were recruited as the control group. 

Data indicate no significant change in CD4/CD8 ratio (CD4/CD8 ratio EMS vs. CD4/CD8 ratio EMS-MS, *p* = 0.62) or in B cell level (B cells EMS vs. B cells EMS-MS, *p* = 0.62). The two groups did not differ in NK effectors (NK EMS vs. NK EMS-MS, *p* = 0.8) or in iNKT lymphocyte levels (iNKT EMS vs. iNKT EMS-MS, *p* = 0.65) (Figure 2).

The activation level of the adaptive immunity effectors (CD4^+^ and CD8^+^ T lymphocytes) was evaluated through CD54 molecule expression. As shown (Figure 3), the CD54 level was significantly increased only in helper T cells (CD54 on CD4 EMS vs. CD54 on CD4 EMS-MS, *p* <0.05) (Figure 3A), while no significant difference was observed in the cytotoxic lymphocytes (CD54 on CD8 EMS vs. CD54 on CD8 EMS-MS, *p* = 0.13) (Figure 3B).

As shown (Figure 4), patients affected by EMS with MS show an increase in Treg subset (Figure 4A) (Foxp3 CD4 EMS vs. Foxp3 CD4 EMS-MS, *p* = 0.32); a higher expression trend was also observed in the exon 2 Foxp3 transcription factor (Figure 4B) (Foxp3 CD4 ex2 EMS vs. Foxp3 CD4 ex2 EMS-MS, *p* = 0.27), largely associated with effective immune-modulating properties of the Treg population. 

The immunological results did not reach statistical significance, except for CD54 expression on CD4 cells. The increased CD4/CD8 ratio and circulating Treg percentage with reduced B cells and CD54 expression on CD8 T cells can be considered a trend that needs further investigation.

## 4. Discussion

Although MS and EMS are clearly different diseases, several studies seem to highlight a common pathogenetic background [8,11]. In our study, we tried to evaluate the possible MS–EMS association in an Italian cohort population. 

Considering the first objective, our study is the first Italian one investigating the prevalence of MS in women with EMS, through the study of the endometriosis population of our referral center. We found a higher prevalence of MS in patients with EMS compared to the MS prevalence in the general Italian population (113/100,00) [14]; indeed, if the prevalence of women with MS in the general female population is estimated to be 80:100,000, our data suggest that the prevalence of MS in patients with EMS is 500:100,000. 

From the analysis of the clinical data, both conditions were found take a mild form; none of the patients had severe neurological disability and none of the patients needed surgical–pharmacological intervention for EMS. 

Regarding the fertility aspect, just three patients desired pregnancy and/or had a baby, while the others had not expressed the desire of a pregnancy at the time of the study. 

The diagnosis of EMS was found to follow that of MS in the whole patient group and in most cases, the diagnosis of EMS was fortuitous. This clinical evidence could be justified by the fact that, as described in the literature, the simultaneous appearance of multiple autoimmune diseases makes the severity and age of onset of symptoms milder [15].

From a purely speculative perspective, these findings leave space for the hypothesis that the therapy for MS has in some way also modulated the inflammatory activity of EMS; however, given the heterogeneity of MS therapies and the small group of patients, it is not possible to establish whether one of these have a better impact on EMS than another. 

In relation to the autoimmune pattern, two patients were diagnosed with Hashimoto’s thyroiditis, demonstrating how the coexistence of multiple autoimmune diseases can be a distinctive element in this selected population [16].

Accumulated evidence demonstrated the involvement of immunological dysregulation in the pathogenesis of both MS and EMS [17,18,19]. The increasing CD4/CD8 ratio and circulating Treg percentage with reduced B cells and CD54 expression on T cells observed in the present study need further investigation.

A limitation of our study is the small number of women enrolled in each group. In a retrospectively studied tertiary endometriosis center, the prevalence was 9 out of 1652 (545/100,000) When compared with the prevalence in the general population of 80 per 100,000 [14], the increase may be due to the selection bias of tertiary care and the retrospective nature of this study [20,21,22]. 

## 5. Conclusions

Considering the complexity of the diseases and the possible immunological mechanisms that could be shared by these two diseases, clinicians caring for adolescents with MS should be aware that it could be associated with EMS, especially if they complain of pelvic pain or infertility; at the same time, patients with EMS should be monitored closely to ensure the early detection of MS. Starting from the new theory of the autoimmune pathogenesis of EMS, the presence of one autoimmune disease should alert one to watch for another. In many cases, the presence of one autoimmune disorder helps lead to the discovery of other autoimmune conditions. To validate the results, further studies with large-scale prospective studies are clearly warranted. 

## Figures and Tables

**Figure 1 jcm-12-02043-f001:**
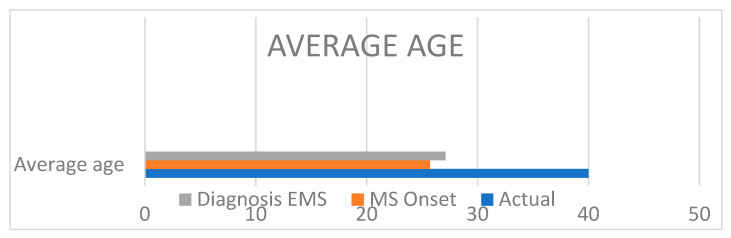
The onset diagnosis of MS anticipates, on average, that of endometriosis.

**Figure 2 jcm-12-02043-f002:**
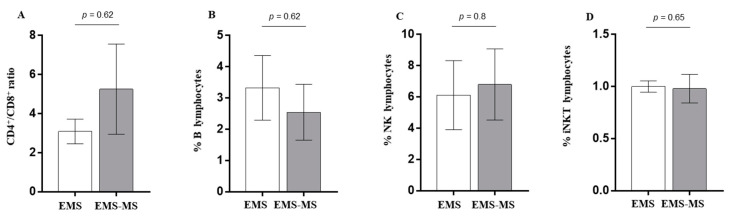
Immune profile of patients with endometriosis (group 1, white column; N = 7) or affected by endometriosis associated with multiple sclerosis (group 2, gray column; N = 7). As shown, we observed an increased ratio of CD4 T versus CD8 T lymphocytes (CD4/CD8 ratio EMS vs. CD4/CD8 ratio EMS-MS, *p* = 0.62) (**A**) and a decreased level of B cells (B cells EMS vs. B cells EMS-MS, *p* = 0.62) (**B**). A similar percentage of NK effectors (NK EMS vs. NK EMS-MS, *p* = 0.8) and of NTK lymphocytes (iNKT EMS vs. iNKT EMS-MS, *p* = 0.65) were also detected (**C**,**D**).

**Figure 3 jcm-12-02043-f003:**
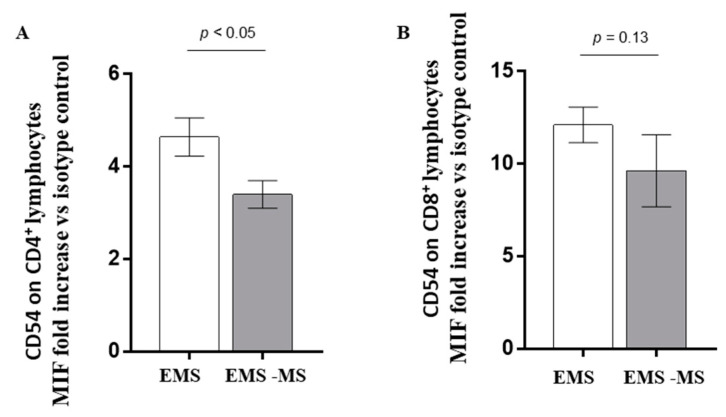
The activation level of the adaptive immunity effectors (CD4^+^ and CD8^+^ T lymphocytes) was evaluated through CD54 molecule expression. As shown, the CD54 level is increased in both helper (CD54 on CD4 EMS vs. CD54 on CD4 EMS-MS, *p* < 0.05) (**A**) and cytotoxic (CD54 on CD8 EMS vs. CD54 on CD8 EMS-MS, *p* = 0.13) (**B**) T cells in the group of women with endometriosis not affected by MS (white column).

**Figure 4 jcm-12-02043-f004:**
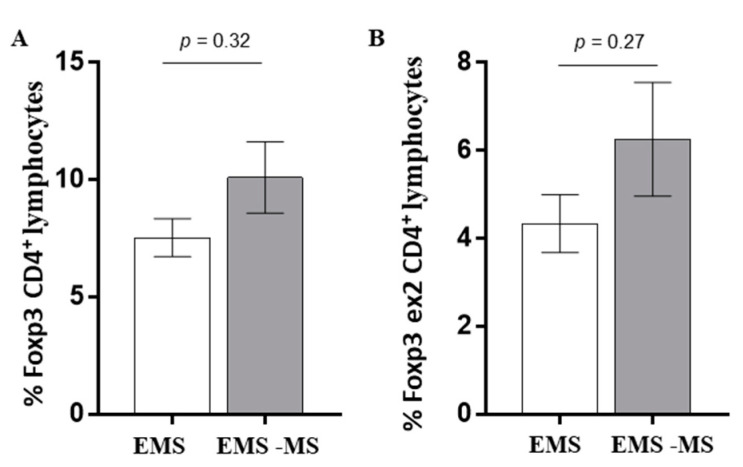
Patients affected by endometriosis with multiple sclerosis (gray column) show increased levels of Treg subset with a higher expression of exon 2 Foxp3 transcription factor, compared with patients with endometriosis (white column) (Foxp3 CD4 EMS vs. Foxp3 CD4 EMS-MS, *p* = 0.32) (**A**). Foxp3 Exon2 expression analysis confirmed such observation (Foxp3 CD4 ex2 EMS vs. Foxp3 CD4 ex2 EMS-MS, *p* = 0.27) (**B**).

**Table 1 jcm-12-02043-t001:** General and endometriosis-related characteristics of control group patients (ND = not desired).

Patient	Actual Age	Age at Diagnosis	Bmi	Dysmenorrhea	Menses	Stage (afs)	Surgery	Therapy	Pregnancy
1	40	20	26	YES	Regular	3	NO	Dienogest (2019)	ND
2	32	23	24	NO	Regular	4	YES	NO	NO
3	30	18	25	NO	Regular	4	NO	Dienogest (2019)	YES
4	36	20	26	YES	Regular	3	NO	NO	YES
5	42	25	27	YES	Regular	3	NO	NO	NO
6	40	22	24	NO	Regular	3	NO	NO	ND
7	39	34	26	NO	Regular	3	NO	NO	ND
8	42	35	25	YES	Regular	4	NO	NO	ND

**Table 2 jcm-12-02043-t002:** General characteristics of patients (ND = not desired).

Patient	Actual Age	BMI	Pregnancy
1	43	27	ND
2	28	26	YES
3	28	28	ND
4	46	26	YES
5	36	28	YES
6	48	25	ND
7	47	26	ND
8	42	25	ND
9	35	26	YES

**Table 3 jcm-12-02043-t003:** Clinical and therapeutic characteristics of endometriosis (For pain evaluation, an NRS scale was used; it goes from “0”: no pain, up to “10”: worst pain).

Patient	Age at Diagnosis	Dysmenorrhea	Pain (Nrs Scale)	Menses	Usg	Stage (afs)	Surgery	Therapy
1	28	YES	3	Regular	Endometrioma	3	NO	NO
2	22	YES	3	Regular	Endometrioma	3	NO	NO
3	20	YES	4	Regular	Endometrioma	3	NO	Dienogest (2016–2018)
4	27	YES	3	Regular	Endometrioma	3	NO	NO
5	21	NO	0	Regular	Endometrioma	3	NO	NO
6	35	NO	0	Oligomenorrhea	Endometrioma	3	NO	NO
7	42	NO	0	Regular	Retrocervical Nodule	4	NO	NO
8	30	NO	0	Regular	Endometrioma+ Lus Nodule	4	NO	NO
9	19	YES	3	Oligomenorrhea	Endometrioma+ Lus Nodule	4	NO	NO

**Table 4 jcm-12-02043-t004:** Clinical and therapeutic characteristics of multiple sclerosis (RR = remitting–relapsing; EDSS = expanded disability status scale).

Patient	Onset Age	Edss	Type	Therapy
1	27	4.5	RR	Ocrelizumab
2	24	3.5	RR	Natalizumab
3	21	1.5	RR	Fingolimod
4	25	2.5	RR	Interferon Beta
5	23	3	RR	Glatiramer
6	31	2	RR	Glatiramer
7	23	6	RR	Dimethyl Fumarate
8	26	2.5	RR	Glatiramer
9	32	1.5	RR	Glatiramer

## Data Availability

The data used to support the findings of this study are included within the article.

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
