# Peer review of "Epidemiological and Immune Profile Analysis of Italian Subjects with Endometriosis and Multiple Sclerosis"

_jcm, 2023, doi:10.3390/jcm12052043_

Round 1

Reviewer 1 Report

The first study investigating the prevalence of MS in women with EMS n the Italian general population (113/100.00), comparatively with the general female population is very interesing and useful for other investigations.

The aim of this manuscript is to estimate the coincidence and compare immune profile of the endometriosis with rare diseases multiple sclerosis, in Italian patients.

Authors have been investigated 1652 patients with endometriosis and have postulated thir vies about potential link between EMS and MS, due to autoimmune peritoneal environment.

These findings may be useful for other researchers.

Author Response

Thank you for reviewing our manuscript and for your comments. We are glad to read about your point of view. 

Reviewer 2 Report

Zizolfi et al. perform an epidemiological analysis, clinical profile, and immune profile analysis of an Italian cohort of patients with endometriosis (EMS) and multiple sclerosis (MS). They report an increased prevalence of MS in women with EMS compared to the general population and non-statistically significant changes in circulating immune cell populations between EMS only patients and EMS-MS patients. The question of whether EMS and MS are biologically associated is a clinically important question, and the sample size for the epidemiological analysis is a strength in this study. The sample size for immune profile analysis, however, is very small, there are significant questions about the composition of the control group, and the data analysis is incomplete. Therefore, conclusions made by the authors go beyond the evidence, and substantial work is needed before this study is ready for publication.

 General comments:

1.     The Methods do not explain how the database from which the patients were recruited was initially formed. Understanding the population that the study sub-population was drawn from is critical to evaluating potential sampling bias. Additionally, no assessment was reported as to whether the study sub-population was comparable to the general Italian population to which the epidemiological comparison was made.

2.     The EMS control group for immune profile analysis is not sufficiently described. At a minimum, the same categories of patient characteristics that were reported in the patient group in Tables 1-2 should be reported for the controls along with an assessment of the comparability of the two groups.

3.     Comparative statistics are reported for the data in the figures but rather a simple statement that no comparisons reached statistical significance. P values should be reported. Furthermore, the error bars in the graphs are not defined. This makes the “trends” in the data impossible to evaluate.

4.     No statistical risk assessment is made comparing the prevalence of MS in patients with EMS to the general Italian population. This makes the conclusion that the prevalence of MS is higher in EMS patients impossible to evaluate. Without statistically significant differences in the epidemiologic, clinical, or immunologic arms of the study, the conclusion that “these results can be taken as a warning for clinicians” (lines 192-193) is not warranted by the data.

Specific comments:

1.     The introduction states the elevated risk of MS in EMS patients as uncontested fact based on a single study (Sinaii et al., 2002; line 40-42), but the literature is not all in agreement. The authors should also mention previously published evidence that does not support this conclusion (e.g. Nielsen et al., 2011, PMID: 21471158).

2.     The legend in Table 4 mistakenly states “SM Onset” rather than “MS Onset”.

3.     In Figure 1, Panel C and D are not labeled in the figure as in the legend.

4.     In Figure 2, the Y-axis labels are identical for A and B. A should report CD4+ cells according to the legend.

5.     There is not enough data presented to comment on the fertility status of the patients as a group as in lines 174-175. The only conclusion that can be drawn is that not all the patients were infertile.

6.     The statement “as described in the literature, the simultaneous appearance of multiple autoimmune diseases, make the severity and age of onset of symptoms milder” (lines 178-179) should include cited references.

Author Response

Reply: Thank you for giving us the chance to enhance our manuscript entitled " Epidemiological and immune profile analysis of Italian subjects with Endometriosis and Multiple Sclerosis"

Below we have listed each question raised, our response, and the position in the paper where issue is mentioned. Pages and numbers refer to marked copy. We also submitted a clean non-edited copy of the revised manuscript.

General comments:

  1. A) Comment #1: The Methods do not explain how the database from which the patients were recruited was initially formed. Understanding the population that the study sub-population was drawn from is critical to evaluating potential sampling bias. Additionally, no assessment was reported as to whether the study sub-population was comparable to the general Italian population to which the epidemiological comparison was made.
  2. B) Response: Thank you for giving us the chance to enhance our manuscript. We deeper explain in the text our methods of patients’ selection.
  3. C) Location: Page 2, lines 64-77.

  1. A) Comment #2: The EMS control group for immune profile analysis is not sufficiently described. At a minimum, the same categories of patient characteristics that were reported in the patient group in Tables 1-2 should be reported for the controls along with an assessment of the comparability of the two groups.
  2. B) Response: Thank you for the suggestion. We add a Table 1 with all the information required.
  3. C) Location: Page 3, Table 1.

  1. A) Comment #3: Comparative statistics are reported for the data in the figures but rather a simple statement that no comparisons reached statistical significance. P values should be reported. Furthermore, the error bars in the graphs are not defined. This makes the “trends” in the data impossible to evaluate.
  2. B) Response: Thank you for the suggestion. We modify the text and add the Pvalue.
  3. C) Location: Pages 7-8, lines 167-171, 176-179, 183-184, 188-189, 197-199, 203-205.

  1. A) Comment #4: No statistical risk assessment is made comparing the prevalence of MS in patients with EMS to the general Italian population. This makes the conclusion that the prevalence of MS is higher in EMS patients impossible to evaluate. Without statistically significant differences in the epidemiologic, clinical, or immunologic arms of the study, the conclusion that “these results can be taken as a warning for clinicians” (lines 192-193) is not warranted by the data.
  2. B) Response: Thank you for the advice. We understand that as written it could seems not supported by data, so we modify the text.
  3. C) Location: Page 9, lines 228-256

Specific comments:

A)Comment #1: The introduction states the elevated risk of MS in EMS patients as uncontested fact based on a single study (Sinaii et al., 2002; line 40-42), but the literature is not all in agreement. The authors should also mention previously published evidence that does not support this conclusion (e.g. Nielsen et al., 2011, PMID: 21471158).

  1. B) Response: Thank you for the suggestion. We add this consideration in the text.
  2. C) Location: Page 2, lines 42-46

  1. A) Comment #2: The legend in Table 4 mistakenly states “SM Onset” rather than “MS Onset”.
  2. B) Response: Thank you for the advice. We modify it.
  3. C) Location: Page 6, Table 5* (* it is now five, since the previous comments reply)

  1. A) Comment #3: The legend in Table 4 mistakenly states “SM Onset” rather than “MS Onset”.
  2. B) Response: Thank you for the advice. We modify it.
  3. C) Location: Page 6, Figure 1.

  1. A) Comment #4:  In Figure 2, the Y-axis labels are identical for A and B. A should report CD4+ cells according to the legend.
  2. B) Response: Thank you for the advice. We modify it.
  3. C) Location: Page 7, Figure 2.

  1. A) Comment #5:  There is not enough data presented to comment on the fertility status of the patients as a group as in lines 174-175. The only conclusion that can be drawn is that not all the patients were infertile.
  2. B) Response: Thank you for the advice. We modify it.
  3. C) Location: Page 9, lines 225-226.

  1. A) Comment #6:  The statement “as described in the literature, the simultaneous appearance of multiple autoimmune diseases, make the severity and age of onset of symptoms milder” (lines 178-179) should include cited references.
  2. B) Response: Thank you for the advice. We add it.
  3. C) Location: Page 9, lines 230.

Reviewer 3 Report

Thank you for this interesting review investigating the association between MS and endometriosis. Please explain why all patients are described to have mild forms of both diseases (lign 100) when table 2 clearly states that all patientins in question have stage 3 or 4 disease at least suffering from an endometrioma. As it is widely known now that AFS classification does not really help in assessing severity of the disease another classification system should be used. I would suggest the ENZIAN or #ENZIAN classification. Otherwise severity would need to be described in more detail (pain on an NRS scale, presence of deep endometriosis, ...). Please also explain, why almost all of these patients who suffer from endometriosis being diagnosed with an endometrioma and at least partially trying to become pregnant did neither receive surgery nor ART or if no child wish was imminent were not treated medically. This might not be relevant to the question of association between endometriosis and MS but draws into question the management of the disease at your facility. Also please clarify what "EDSS" in table 3 stands for. This abbreviation is not explained anywhere in the article if I am not mistaken. Generally all abbreviations need to be introduced. In case of tables and figures these abbrevations should additionally be explained in the subheadings. Please clarify what exactly you describe in table 4 and why this is relevant. 

Author Response

Reply: Thank you for giving us the chance to enhance our manuscript entitled " Epidemiological and immune profile analysis of Italian subjects with Endometriosis and Multiple Sclerosis"

Below we have listed each question raised, our response, and the position in the paper where issue is mentioned. Pages and numbers refer to marked copy. We also submitted a clean non-edited copy of the revised manuscript.

  1. A) Comment #1: Please explain why all patients are described to have mild forms of both diseases (lign 100) when table 2 clearly states that all patients in question have stage 3 or 4 disease at least suffering from an endometrioma. As it is widely known now that AFS classification does not really help in assessing severity of the disease another classification system should be used. I would suggest the ENZIAN or #ENZIAN classification. Otherwise, severity would need to be described in more detail (pain on an NRS scale, presence of deep endometriosis, ...).
  2. B) Response: Since we refer to a database that began in 2012, a period in which the reference classification was the AFS, to ensure homogeneity of the data, we made the choice to continue using this type of classification. For this reason, considering your suggestion, we will add the pain scale to improve our work.
  3. C) Location: Page 5, Table 3.

  1. A) Comment #2: Please also explain, why almost all of these patients who suffer from endometriosis being diagnosed with an endometrioma and at least partially trying to become pregnant did neither receive surgery nor ART or if no child wish was imminent were not treated medically. This might not be relevant to the question of association between endometriosis and MS but draws into question the management of the disease at your facility.
  2. B) Response: Thanks for this comment. Considering the symptoms (summarized in table 3), patients complaining mild dysmenorrhea (NRS 3/10) reported to use FANS on demand to control pain; the only patient complaining moderate dysmenorrhea (NRS 6/10) has been treated using hormonal therapy;
  3. C) Location:-

  1. A) Comment #3: Also please clarify what "EDSS" in table 3 stands for. This abbreviation is not explained anywhere in the article if I am not mistaken. Generally, all abbreviations need to be introduced. In case of tables and figures these abbreviations should additionally be explained in the subheadings.
  2. B) Response: At page 2, line 70 we introduce for the first time the abbreviation EDSS and we take care of adding what It stands for (Expanded disability status scale). However, we accept your suggestion and write the extended form also in the table.
  3. C) Location: Page 6, lines 150.

  1. A) Comment #4: Please clarify what exactly you describe in table 4 and why this is relevant.
  2. B) Response: This table has been formulated to observe the temporal characteristics of the onset of the two diseases: we wanted to add the data of which of the two diseases was diagnosed first and the temporal distance between the diagnosis of the two conditions.
  3. C) Location:-

Round 2

Reviewer 2 Report

In the revised manuscript, the authors partially but insufficiently addressed my concerns.

From general comment 1, the methods were appropriately updated, but there is still no assessment of the study subpopulation compared to the general Italian population or even to other endometriosis patient populations from other reports from other clinics. This is important so that the possibility of a biased or unusual patient cohort can be assessed.

From general comment 2, individual patient data are now listed for EMS controls in Table 1, but neither descriptive statistics nor statistical tests comparing control group characteristics to the EMS-MS group are reported. These are critical to assess the comparability of the groups.

From general comment 3, statistical test results have been added appropriately to the figures, but the corresponding changes in the interpretation are inconsistent. The revision changes the interpretation of the CD4/CD8 ratio to no significant difference between ENDO and ENDO-MS groups but still indicates a decrease in the %B lymphocytes even though it does not reach statistical significance (Line 171-172; Fig 1). Additionally, the CD54 level is stated to be increased in both CD4 and CD8 T cells of the ENDO group, but only the CD4 cell data passes the statistical significance test (Lines 186-190; Fig 2). Finally, %Foxp3 and Foxp3 ex2 are still stated to be higher in the EMS-MS group despite no statistical significance (Lines 201-205; Fig 3). In this section, the abbreviations for the groups have also inexplicably changed from “EMS” and “EMS-MS” to “ENDO” and “ENDO-MS”, which is confusing to the reader.

From specific comment 2, the mistaken use of “SM” in place of “MS” was corrected in the revision Table 5, but the same mistake was also made in the revision Table 4 and in the text lines 172-175, 181-189, 195-196, 203-205, 210-211.

Author Response

Reply: Thank you for the suggestions. Below we have listed each question raised, our response, and the position in the paper where issue is mentioned. Pages and numbers refer to marked copy. We also submitted a clean non-edited copy of the revised manuscript.

General comments:

  1. A) Comment #1: The methods were appropriately updated, but there is still no assessment of the study subpopulation compared to the general Italian population or even to other endometriosis patient populations from other reports from other clinics. This is important so that the possibility of a biased or unusual patient cohort can be assessed.
  2. B) Response: The EMS-MS group was recruited from a general database of EMS patients from our clinic. We can assume that this database reflects the general Italian population because there is no other point of commonality among these women other than endometriosis. Therefore, we limited ourselves to looking in the anamnesis of these women if they had reported other pathologies and if MS was found they were recruited.
  3. C) Location: -

  1. A) Comment #2: Individual patient data are now listed for EMS controls in Table 1, but neither descriptive statistics nor statistical tests comparing control group characteristics to the EMS-MS group are reported. These are critical to assess the comparability of the groups.
  2. B) Response: Thank you for the comment. We perform a descriptive statistics to compare the two groups. This is the table showing the matching of the two groups.

A AGE

D AGE

BMI

DYSMENORREA

MENSES

STAGE

SURGERY

THERAPHY

PREGNANCY

C

S

C

S

C

S

C

S

C

S

C

S

C

S

C

S

C

S

1

40

43

20

28

26

27

YES

YES

REGULAR

REGULAR

3

3

NO

NO

YES

NO

ND

ND

2

32

28

23

22

24

26

NO

YES

REGULAR

REGULAR

4

3

YES

NO

NO

NO

NO

YES

3

30

28

18

20

25

28

NO

YES

REGULAR

REGULAR

4

3

NO

NO

YES

YES

YES

ND

4

36

46

20

27

26

26

YES

YES

REGULAR

REGULAR

3

3

NO

NO

NO

NO

YES

YES

5

42

36

25

21

27

28

YES

NO

REGULAR

REGULAR

3

3

NO

NO

NO

NO

NO

YES

6

40

48

22

35

24

25

NO

NO

REGULAR

NOT REGULAR

3

3

NO

NO

NO

NO

ND

ND

7

39

47

34

42

26

26

NO

NO

REGULAR

REGULAR

3

4

NO

NO

NO

NO

ND

ND

8

42

42

35

30

25

25

YES

NO

REGULAR

REGULAR

4

4

NO

NO

NO

NO

ND

ND

MEAN

37.6

42.5

27.5

29

25.5

26

4 YES

4 NO

4 YES

4 NO

100% REGULAR

86% REGULAR

5 STAGE 3

3 STAGE 4

6 STAGE 3

2 STAGE 4

86% NO

100% NO

75% NO

86%

NO

2 PREGNANCY

3 PREGNANCY

In this table we match the 8 patients of the control group and the 8 patients with EMS-MS who did blood sample. (C: CONTROL GROUP (EMS), S: STUDY GROUP (EMS-MS), A.AGE: ACTUAL AGE, D.AGE: AGE AT DIAGNOSIS OF EMS, ND: NOT DESIRED).

  1. C) Location: Page 3, lines 99-100.

  1. A) Comment #3: statistical test results have been added appropriately to the figures, but the CD4/CD8 ratio to no significant difference between ENDO and ENDO-MS groups but still indicates a decrease in the %B lymphocytes even though it does not reach statistical significance (Line 171-172; Fig 1). Additionally, the CD54 level is stated to be increased in both CD4 and CD8 T cells of the ENDO group, but only the CD4 cell data passes the statistical significance test (Lines 186-190; Fig 2). Finally, %Foxp3 and Foxp3 ex2 are still stated to be higher in the EMS-MS group despite no statistical significance (Lines 201-205; Fig 3). In this section, the abbreviations for the groups have also inexplicably changed from “EMS” and “EMS-MS” to “ENDO” and “ENDO-MS”, which is confusing to the reader.
  2. B) Response: Thank you for this suggestion. We update the text with the correction
  3. C) Location: Pages 7-8, paragraph 3.3 (highlight in yellow).

  1. A) Comment #4: changes in the interpretation are inconsistent. The revision changes the interpretation of the from specific comment 2, the mistaken use of “SM” in place of “MS” was corrected in the revision Table 5, but the same mistake was also made in the revision Table 4 and in the text lines 172-175, 181-189, 195-196, 203-205, 210-211.
  2. B) Response: We are sorry for this mistake. We correct it throughout the text.
  3. C) Location:-
